# LatentWarp: Consistent Diffusion Latents for Zero-Shot Video-to-Video Translation

## Abstract

Leveraging the generative ability of image diffusion models offers great potential for zero-shot video-to-video translation. The key lies in how to maintain temporal consistency across generated video frames by image diffusion models. Previous methods typically adopt cross-frame attention, *i.e.,* sharing the *key* and *value* tokens across attentions of different frames, to encourage the temporal consistency. However, in those works, temporal inconsistency issue may not be thoroughly solved, rendering the fidelity of generated videos limited. In this paper, we find the bottleneck lies in the unconstrained query tokens and propose a new zero-shot video-to-video translation framework, named *LatentWarp*. Our approach is simple: to constrain the query tokens to be temporally consistent, we further incorporate a warping operation in the latent space to constrain the query tokens. Specifically, based on the optical flow obtained from the original video, we warp the generated latent features of last frame to align with the current frame during the denoising process. As a result, the corresponding regions across the adjacent frames can share closely-related query tokens and attention outputs, which can further improve latent-level consistency to enhance visual temporal coherence of generated videos. Extensive experiment results demonstrate the superiority of *LatentWarp* in achieving video-to-video translation with temporal coherence.

## 1 Introduction

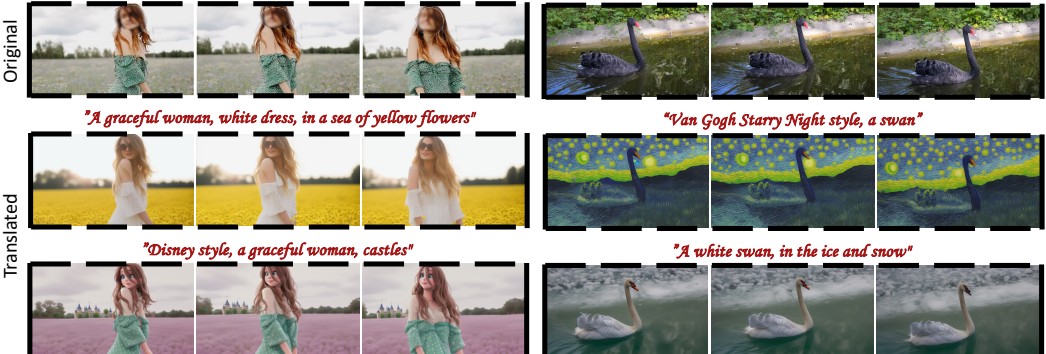

Figure 1: *LatentWarp*, a new zero-shot video-to-video translation framework, possesses the capability of performing high-quality video translation with temporal consistency. By supplying the original video and a target text prompt, *LatentWarp* empowers users to seamlessly translate diverse videos with the target style, while maintaining the temporal coherence of the original video content.

The recent advancements in large-scale text-to-image generative models Rombach et al. (2022); Nichol et al. (2021); Ho et al. (2022a) have demonstrated impressive capabilities in content creation and editing, empowering users to generate images in various styles by providing corresponding prompts. However, transferring the style of videos while maintaining temporal coherence remains a significant challenge. Existing diffusion-based methods have struggled with maintaining consistency across frames, resulting in diverse and uncontrollable generated results, as well as visual artifacts and flickering during video playback. To address this challenge, recent research efforts Singer et al. (2022); Esser et al. (2023) have attempted to collect large-scale text-video paired datasets to

train video diffusion models. However, the style of the collected videos often appears more tedious compared to image datasets, making it challenging to achieve high-quality video-to-video translation. Therefore, as a promising research direction, leveraging the generative ability of image diffusion models, such as Stable Diffusion Rombach et al. (2022), for zero-shot video-to-video translation of arbitrary styles offers great potential. By extending image diffusion models to handle video data, it may be possible to achieve more controlled and coherent video generation, enabling users to translate videos into different styles without the need for paired training data.

The main challenge in zero-shot video translation is maintaining temporal consistency across frames. When translating videos frame-by-frame independently, the diversity of the diffusion process can lead to the generation of various types of frames, resulting in flickering and inconsecutive videos. The state-of-the-arts Wu et al. (2022); Khachatryan et al. (2023); Ceylan et al. (2023) have introduced a temporal attention mechanism, by replacing or extending the self-attention layers of diffusion models with cross-frame attention. However, we observe that although the global appearance is improved compared to generating each frame independently, there still remains inconsistency in the details across frames as illustrated in Fig. 2. We investigate into this phenomenon and would attribute this inconsistency to the variation in *query* tokens across different frames. The reason is the *key* and *value* tokens are shared and fixed in cross-frame attention, but the *query* tokens are adopted from the current frame during the attention operation. Unconstrained *query* tokens would result in inconsistent attention outputs, further leading to variation in latent feature and pixel values. Logically, if corresponding regions in each frame share the same *query* tokens along with the shared *key* and *value* tokens, these regions

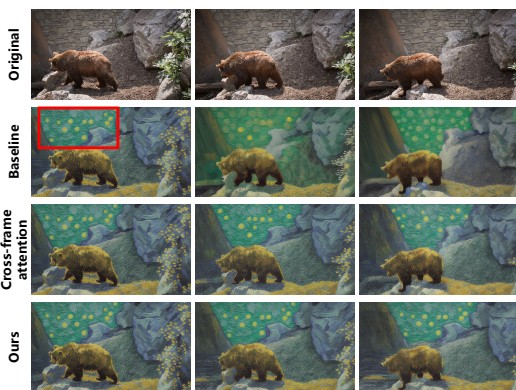

Figure 2: Illustration of the respective effects of cross-frame attention which constrains the *key* and *value* tokens to be the same across frames, and our method that additionally constrains the *query* tokens by warping the latents. The baseline refers to independently translating each frame with ControlNet. By examining the pattern of randomly generated stars in the background, we observe that cross-frame attention only constrains the global style or appearance, while the details change across the entire sequence. In contrast, our method effectively maintains temporal consistency, ensuring both global and detailed consistency throughout the video.

would receive the same attention results and consistent latent features, thus keeping consistent visual details across frames.

In this paper, we propose a novel framework for zero-shot video-to-video translation that aims to address the aforementioned issues and enhances the temporal consistency of translated videos. Specifically, we save and utilize the generated latents from previous frames during the denoising process, and then apply a warping operation on the saved latents with the down-sampled optical flow maps obtained from the original video. The latents of the last frame are aligned with the current frame through this process. The advantage of our proposed framework is that the corresponding regions share the same *query*, *key*, *value* tokens, resulting in consistent attention outputs. By applying the warping operation and obtaining *query* tokens from the latent features, we ensure that the corresponding regions between adjacent frames have closely related *query* tokens, in addition to the shared *key* and *value* tokens provided by cross-frame attention. As a result, it could be guaranteed that the latent representations of corresponding areas between adjacent frames are generated closely, thus enhancing the temporal coherence of the generated videos.

In a nut shell, our contributions can be summarized as follows:

- We propose a novel zero-shot video-to-video translation framework, *LatentWarp*. By warping the latent features from the previous frame to align with the current frame using optical flow maps, we enhance the visual consistency of the translated videos.

- Our proposed framework imposes constraints on the *query* tokens during the attention process, which has been ignored in previous works. Complementary to shared *key* and *value*

- tokens, our approach encourages corresponding regions of adjacent frames to receive similar attention outputs and latent features.
- Experimental results demonstrate that our *LatentWarp* can achieve state-of-the-art temporal consistency and higher quality on translated videos, especially for complicated motions.

## 2 RELATED WORKS

### 2.1 DIFFUSION MODELS FOR IMAGE GENERATION

Diffusion model Ho et al. (2020); Rombach et al. (2022)has demonstrated remarkable performance in generating high-quality images, surpassing the previous approaches such as Generative Adversarial Networks (GANs) Goodfellow et al. (2020). Attributed to the availability of large-scale text-image paired datasets for training the text guidance, users can generate realistic and diverse images by providing simple text inputs. GLIDE Nichol et al. (2021) introduced text conditions to the diffusion model and leveraged classifier information to enhance the quality of generated images. DALLE-2 Ramesh et al. (2022) further enhances performance by utilizing the joint feature space of CLIP Radford et al. (2021) as a condition. Imagen Saharia et al. (2022) achieves photorealistic image generation by leveraging the strength of both the language and a cascade of diffusion models. Latent diffusion model Rombach et al. (2022) performs the denoising process in the latent space of an auto-encoder to reduce the computational resources.

Unlike the aforementioned works that share parameters during the generation process, some recent works aim to enhance the control and manipulation capabilities of text-to-image (T2I) models. ControlNet Zhang & Agrawala (2023) allows users to influence specific attributes or properties of the generated images, such as controlling the pose, depth map, or other visual factors.

The compelling image quality generated by diffusion model has advanced the field of T2I generation and opened up new possibilities for video generation. However, T2I models are primarily designed for generating single images, and applying them to the video domain necessitates additional considerations to maintain temporal coherence and consistency across frames.

### 2.2 DIFFUSION MODELS FOR VIDEO GENERATION AND TRANSLATION

Recently, several methods based on diffusion models have been proposed to extend text-to-image models to generate coherent and consistent video sequences. Some of methods show promise in text-to-video generation and typically require large-scale video datasets for training. By training on both image and video data, Video Diffusion Model Ho et al. (2022b) leverages the diffusion process to generate each frame while considering temporal dependencies. Imagen Video Ho et al. (2022a) improves upon the Video Diffusion Model by incorporating a cascade of spatial and temporal video super-resolution models. Make-A-Video Singer et al. (2022) takes an unsupervised learning approach to learn temporal dynamics from video data. Although training on large-scale video datasets allows the models to lead to more realistic and coherent video generation, collecting and curating such large-scale video datasets can be challenging and time-consuming. Tune-A-Video Wu et al. (2022) proposes an efficient attention approach and employs fine-tuning on a single video to generate coherent video sequences.

Compared to above methods, zero-shot video generation and translation with diffusion models offers a flexible and powerful approach for generating videos in different domains without the need for extensive labeled training data. Image models are primarily designed to process individual frames without considering the temporal dependencies between consecutive frames. Therefore, generating consistent videos using image models can be challenging due to the temporal nature of videos. To address this challenge, FateZero Qi et al. (2023) utilizes a blending technique with attention features to ensure temporal consistency and preserving high-level styles and shapes. Text2Video-Zero Khachatryan et al. (2023) directly translates latent representations to simulate motions in the generated videos.Furthermore, Rerender-A-Video Yang et al. (2023) and VideoControlNet Hu & Xu (2023) both focus on achieving pixel-level temporal consistency by introducing a pixel-aware cross-frame latent fusion and motion information, respectively. However, cross-frame attention and pixel-level temporal consistency still have limitations in preserving fine-grained temporal coherence in terms of texture and detailed visual elements.

Different from above methods, our video-to-video translation framework emphasizes latent-level temporal consistency and imposes explicit constraint on the latent space to generate frames that are consistent with the overall style and content of the input video. By incorporating these constraints, the T2I models can produce visually appealing and coherent videos that adhere to the desired style, content, and temporal consistency.

## 3 PRELIMINARIES

### 3.1 DIFFUSION MODELS

Stable diffusion Rombach et al. (2022) is a notable diffusion model that operates within the latent space, which leverages a pretrained autoencoder to map images into latents and reconstruct the latents into high-resolution images, respectively. Starting from an initial input signal $z_0$, which is the latent of an input image $I_0$, the process iteratively progresses through a series of steps. The diffusion forward process is defined as:

$$q\left(z_t \mid z_{t-1}\right) = \mathcal{N}\left(z_t; \sqrt{1 - \beta_{t-1}} z_{t-1}, \beta_t I\right), \quad t = 1, \ldots, T \tag{1}$$

where $q\left(z_t \mid z_{t-1}\right)$ is the conditional density of $z_t$ given $z_{t-1}$, and $\beta_t$ is hyperparameters. $T$ is the total timestep of the diffusion process. The objective of the diffusion model is to learn the reverse process of diffusion, which is commonly used for denoising process. Given a noise $z_t$, the model predicts the added noise at the previous timestep $z_{t-1}$, iteratively moving backward until reaching the original signal $z_0$.

$$p_\theta\left(z_{t-1} \mid z_t\right) = \mathcal{N}\left(z_{t-1}; \mu_\theta\left(z_t, t\right), \Sigma_\theta\left(z_t, t\right)\right), \quad t = T, \ldots, 1 \tag{2}$$

where $\theta$ is the learnable parameters, trained for the inverse process. In Stable Diffusion, the model can be interpreted as a sequence of weight-sharing denoising autoencoders $\epsilon_\theta\left(z_t, t, c_\mathcal{P}\right)$, trained to predict the denoised variant of input $z_t$ and text prompt $c_\mathcal{P}$. The objective can be formulated as:

$$\mathbb{E}_{z, \epsilon \sim \mathcal{N}(0,1), t}\left[\left\|\epsilon - \epsilon_\theta\left(z_t, t, c_\mathcal{P}\right)\right\|_2^2\right] \tag{3}$$

ControlNet Zhang & Agrawala (2023) is a neural network structure for controlling pretrained large diffusion models Ho et al. (2020) by incorporating additional input conditions, such as edge, segmentation, and keypoints inputs, which serves the purpose of providing structure guidance. The combination of large diffusion models and ControlNets empowers the zero-shot video-to-video framework to produce high-quality video outputs with enhanced temporal consistency.

### 3.2 OPTICAL FLOW ESTIMATION

Optical flow estimation is a task to estimate the motion vector of every pixel in a pair of consecutive frames of the video, which captures the apparent displacement of objects between frames. Our proposed LatentWarp utilizes a pre-trained optical flow estimation network to manipulate the motion within the video sequence. Specially, our method aims to modulate and control the motion between corresponding latents of frames, so we directly employ the RAFT Teed & Deng (2020) as our optical flow estimation network in LatentWarp. With the further advancement in optical flow estimation, the progressive approaches, such as Flowformer Huang et al. (2022), would be integrated in our proposed framework to enhance the accuracy and generalizability of optical flow estimation.

### 3.3 ATTENTION MECHANISM

For stable diffusion model, the attention mechanism plays a crucial role in capturing dependencies and guiding the diffusion process, which helps the model focus on pertinent information and selectively propagate it throughout the data.

The self-attention in a stable diffusion model is based on a U-Net Ronneberger et al. (2015) architecture and utilizes the linear projections to derive *query*, *key* and *value* features $Q, K, V \in \mathbb{R}^{h \times w \times c}$ from the input feature. The output of the self-attention layer can be computed by:

$$\text{Self-Attention}(Q, K, V) = \text{Softmax}\left(\frac{QK^T}{\sqrt{c}}\right) V \tag{4}$$

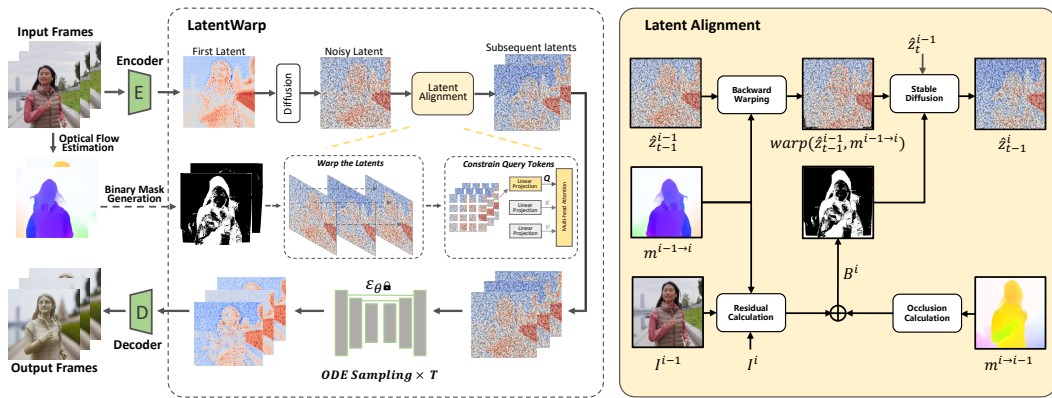

Figure 3: Overview of our proposed framework *LatentWarp*. In the left part of the figure, we show the overall framework of *LatentWarp*. In each denosing step, we warp the latent between adjacent frames to make alignment in the latent space. In the right part, we illustrate the technical details about latent warping, binary mask generation and latent alignment.

In the modified version of Stable Diffusion architecture, each self-attention layer is replaced with a cross-frame attention mechanism. The cross-frame attention operates by attending to the first frame from each subsequent frame in the sequence. Formally,

$$\text{Cross-Frame-Attention}\left(Q^i, K^{1:n}, V^{1:n}\right) = \text{Softmax}\left(\frac{Q^i\left(K^1\right)^T}{\sqrt{c}}\right) V^1 \qquad (5)$$

for $i = 1, \ldots, n$. By incorporating cross frame attention mechanism, the model can generate frames that exhibit greater consistency in terms of appearance, structure, and object identities throughout the video sequence.

## 4 METHODS

In zero-shot video-to-video translation task, given a text prompt $\mathcal{P}$, and an original video $\mathcal{I} = \left[\boldsymbol{I}^1, \ldots, \boldsymbol{I}^n\right]$, where $\boldsymbol{I}^i$ represents the $i$-th frame in the original video, we aim to generate a translated video, $\hat{\mathcal{I}} = \left[\hat{\boldsymbol{I}}^1, \ldots, \hat{\boldsymbol{I}}^n\right]$ that complies with the description of text prompt $\mathcal{P}$. Directly leveraging the generative ability of Stable Diffusion combined with ControlNet could hardly handle the temporal coherence between frames, which would result in visual flickering during video playback. As illustrated in Fig. 3, we propose a new zero-shot video-to-video translation framework, named *LatentWarp*, to tackle the the inter-frame inconsistency issue and ensure smooth transitions between frames. In the following subsections, we would provide a detailed introduction of our proposed methods.

### 4.1 WARP THE LATENTS

As analysed in Sec. 1, unconstrained *query* tokens can result in inconsistent attention outputs, leading to visual inconsistency of generated videos. To address the aforementioned issue and mitigate the inconsistencies caused by unconstrained *query* tokens, we propose warping the latents as a solution. From the original video, we extract the optical flow between adjacent frames, including forward optical flow maps $\mathcal{FM} = \left[m^{1\to0}, \ldots, m^{n\to n-1}\right]$ and backward optical flow maps $\mathcal{BM} = \left[m^{0\to1}, \ldots, m^{n-1\to n}\right]$. We apply a warping operation on the latent of the last frame to align with the current frame using these flow maps. To provide a more direct explanation of our method, we take the $i-1$-th and $i$-th frames for example. In each denoising timestep, we save the $i-1$-th frame's latents, *i.e.*, $\left[\hat{z}_1^{i-1}, \ldots, \hat{z}_T^{i-1}\right]$. Next, by downsampling the backward optical flow maps to match the size of the latent representation, we apply backward warping operation on the saved latents of the last frame and obtain the warped latents, $\left[warp\left(\hat{z}_1^{i-1}, m^{i-1\to i}\right), \ldots, warp\left(\hat{z}_T^{i-1}, m^{i-1\to i}\right)\right]$. Afterwards, the warped latents would be used to align the generation process of the $i$-th frame.

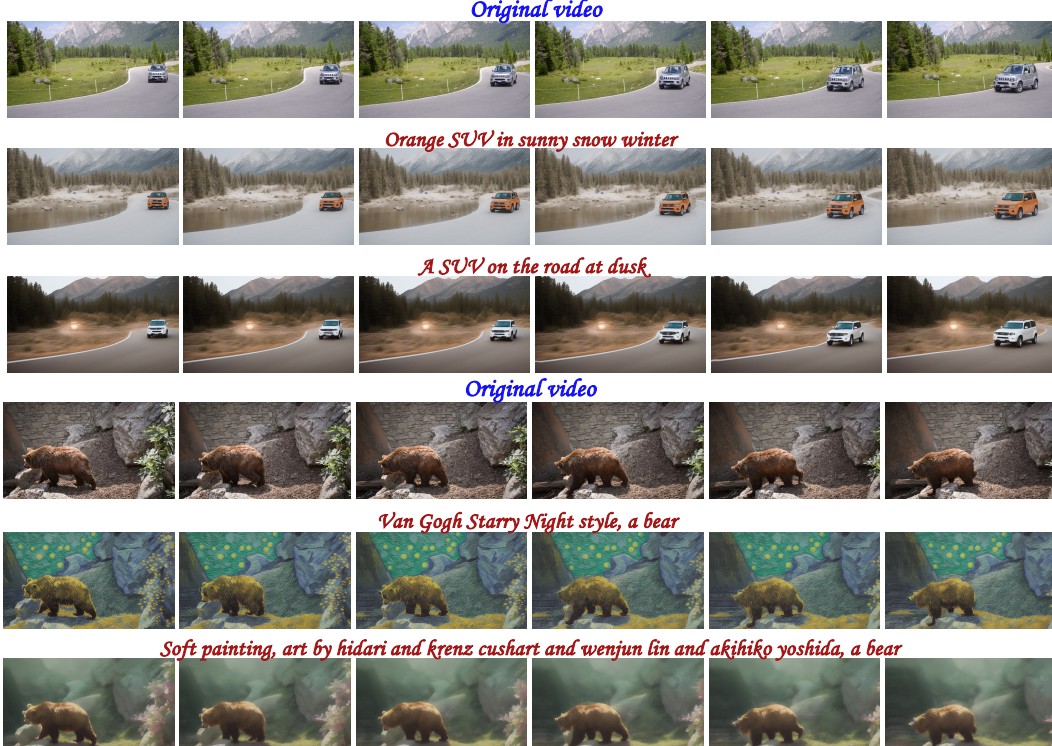

Figure 4: **Results**. The translation results of two videos with different prompts. It could be seen that our method achieves high translation quality combined with strong temporal coherence.

## 4.2 BINARY MASK GENERATION

To ensure alignment between the generation of the $i$-th frame and the warped latent from the $i$-1-th frame, we introduce binary masks $\mathbf{B}$ indicating which regions of the warped latent should be preserved and which regions, *e.g.*, the occlusion regions, should be replaced with latent of the current frame. The generation of binary masks also takes advantages of the dynamic information in the original video. As for the binary mask generation, we refer to a previous work Hu & Xu (2023). The process can be divided into three parts. Firstly, considering the occluded regions $o^i$ in the $i$-th frame, we get $o^i$ by taking forward warping operation on an all-one matrix with the forward optical flow maps $m^{i \rightarrow i-1}$. As a result, the occlusion areas would be zero because its elements have been warped to other regions while no others have been warped to these areas. Although the aforementioned method address the occlusion issue, it could not handle the scenarios of scene switching in videos. Therefore, secondly, we address this problem by calculating the residual map $r^i$ between the backward warping result and the $i$-th frame of the original video $r^i = \left| warp\left(\boldsymbol{I}^{i-1}, m^{i \rightarrow i-1}\right) - \boldsymbol{I}^i \right|$. Finally, taking into consideration of both the occlusion mask $o^i$ and the the residual map $r^i$, we got the binary mask:

$$B^i = \begin{cases} 1 & \text{if } o^i - \alpha r^i > threshold \\ 0 & \text{otherwise} \end{cases} \tag{6}$$

In the mask, 1 denotes keeping the latent warped from the $i-1$-th frame, and 0 denotes adopting the newly generated latent.

## 4.3 LATENT ALIGNMENT

With the warped latents of the $i$-1-th frame and the binary mask $B^i$, we perform latent alignment during the generation process of $\hat{z}^{i-1}$, in which the U-Net output of Stable Diffusion could be formulated as:

$$Output = \epsilon_\theta \left( B^i \cdot warp\left(\hat{z}_t^{i-1}, m^{i-1 \rightarrow i}\right) + \left(1 - B^i\right) \cdot \hat{z}_t^i, t, c_\mathcal{P} \right) \tag{7}$$

Through latent alignment, *query* tokens of temporally corresponding regions are closely generated from related latent feature. Complementary to the shared *key* and *value* tokens provided by cross-

| Methods | CLIP Text (%) ↑ | CLIP Image (%) ↑ | Warp Error ($\times 10^{-3}$) ↓ |
|---|---|---|---|
| Original video | 22.08 | 98.12 | 4.1 |
| Tune-A-Video | 26.78 | 96.74 | 30.1 |
| Text2Video-Zero | 21.54 | 93.17 | 29.8 |
| TokenFlow | 26.43 | 97.53 | 6.1 |
| Ours w/o Cross-Frame-Attn | 28.48 | 97.43 | 6.4 |
| Ours w/o latent alignment | 29.79 | 97.51 | 3.8 |
| Ours | **29.88** | **97.57** | **2.9** |

Table 1: **Quantitative comparison**. To evaluate the quality of translated videos, in terms of prompt alignment and temporal coherence, we calculate "CLIP-Text", average CLIP similarity between text and frames, "CLIP-Image", average CLIP similarity across frames, and warp error. The results show that our method could generate temporal consistent video with high prompt alignment.

frame attention mechanism Khachatryan et al. (2023), the areas would receive closed attention outputs. In this manner, the latent feature would be updated coherently, further preserving inter-frame visual consistency throughout the sequence. In the last few denoising steps, we refrain from performing the latent alignment operation. This is due to the observation of errors introduced in the warping operation and optical flow estimation during these steps, and these errors have a negative impact on the generation quality and result in degraded visual output. Therefore, regular backward process is performed instead, until we get the denoised latent of the $i$-th frame $\hat{z}^i$. The overall algorithm is listed in Algorithm 1. It could be seen that the overall method is simple and concise. In the following section, we would demonstrate the effectiveness of our proposed *LatentWarp* with impressive video translation results.

## 5 EXPERIMENTS

### 5.1 IMPLEMENTATION DETAILS

In implementation, we employ the Stable Diffusion of version 1.5 as the base model. We adopt a kind of ODE sampler, Euler-A Karras et al. (2022), avoiding the stochastic factors introduced by SDE Ho et al. (2020). The number of denoising steps is set to be 20. In the first 16 steps we perform latent alignment while in the last 4 steps we adopt the regular denoising strategy. We utilize ControlNets Zhang & Agrawala (2023) with condition of canny and tile as the frame editing methods. The hyper-parameters, $\alpha$ and $threshold$, are set as 5 and 0.6 for binary mask generation. During the video translation process, we only translate the key frames, while the other frames are propagated from the key frames following the previous methods Yang et al. (2023); Jamriška et al. (2019). The interval between the key frames is set to be 10 or 15. We adopt an off-the-shelf optical estimation networks RAFT Teed & Deng (2020), to obtain the optical flow maps from the original video. The original videos are from a benchmark dataset, DAVIS Perazzi et al. (2016). The spatial resolution of the video is down-sampled to $576 \times 1024$ if its aspect ratio is 9:16, or $512 \times 512$ if the ratio is 1:1. All the experiments are conducted with one NVIDIA A100 GPU.

### 5.2 QUALITATIVE RESULTS

In Fig. 4 we show the results of translated frames. It could be seen that the global scene and background are temporally consistent. Additionally, the identity, appearance, and texture of the foreground object are maintained well throughout the video sequence. In the "car-turn" case, top three rows in Fig. 4, the scenery is changed to a snowy day or midnight. It could be noticed that the snowy peak and the road lamps are temporal consistent throughout the video. In the bottom three rows, the frames are translated to the style of Van Gogh's Starry Night and soft painting. It could be seen that the pattern of randomly generated stars in the background are maintained stably. Moreover, the generated videos are closely aligned with the text prompts. When the given prompts describe the global scene *in a snowy winter* or *at dusk* in the "car-turn" case, the road would be covered with snow with a snowy peak in the background or the street lamp would shed dim light by the road side, creating a realistic atmosphere in the twilight.

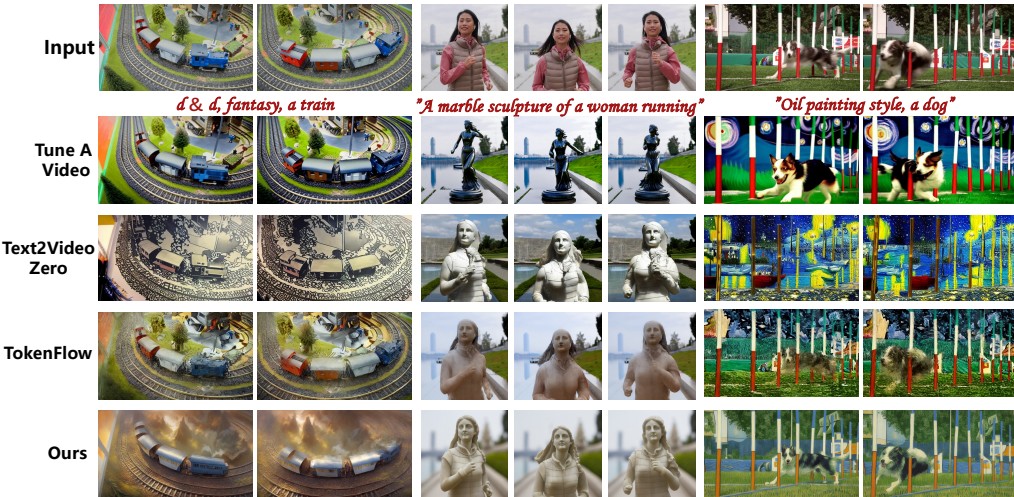

Figure 6: **Qualitative comparison**. We make comparison with the prevailing methods. We re-implement these methods with their officially released code.

### 5.3 COMPARISON WITH PREVIOUS METHODS

We compare our proposed method with the current state-of-the-arts, including: (i)Tune-A-Video Wu et al. (2022), which fine-tunes the inflated attention layers to learn the temporal information of the given video; (ii)Text2Video-Zero Khachatryan et al. (2023), that extends self-attention to cross-frame attention for zero-shot video generation; (iii)TokenFlow Geyer et al. (2023), which propagates the attention tokens according to cosine similarity of the latent feature to maintain inter-frame semantic correspondence. These three methods are popular and representative, which stand for the performance of one-shot tuning, and zero-shot translation through introducing temporal attention or manipulating attention tokens. In the following, we will re-implement these methods with their officially released code and make fair comparison with these methods both quantitatively and qualitatively to demonstrate the superiority of our method.

**Quantitative Comparison**. We expect the translated video to be faithfully aligned with the given prompts while maintaining temporal consistency throughout the entire sequences. Therefore, following previous works Geyer et al. (2023); Ceylan et al. (2023), we adopt the metrics: (i) CLIP score between text prompt and image ("CLIP-Text") which reflects the video-prompt alignment by calculating the average cosine similarity between the CLIP text embedding of the prompt and CLIP image embedding of each frame; (ii) CLIP score between each frame ("CLIP-Image"), which represents the consistency in content-level by measuring the cosine similarity of CLIP image embeddings extracted from consecutive frames; and (iii) "Warp error", which reflects consistency in pixel-level by calculating mean squared error of pixel value between the warped edited frame and the corresponding target frame.

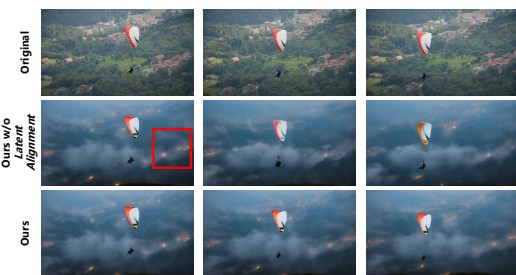

Figure 5: **Ablation study**. We ablate the effect of latent alignment through visualization results. The results show that latent alignment maintains the fine-grained visual details effectively.

The comparison result is provided in Table 1. It should be noticed that our proposed method achieves the best text-video alignment in terms of the highest CLIP-Text score. Additionally, considering our method has the highest CLIP-Image score and the lowest warp error, our framework maintains the temporal coherence of the translated sequence effectively.

**Qualitative Comparison**. As illustrated in Fig. 6, we make qualitative comparisons with these prevailing methods. Compared with Tune-A-Video and Text2Video-Zero, our method exhibits higher

prompt-image alignment while maintaining the motion information from the original video. In comparison with TokenFlow, the results of our method appear more natural and appealing.

## 5.4 Ablation Study

Moreover, we ablate the respective effect of latent alignment and cross-frame attention on maintaining temporal consistency. "Ours w/o latent alignment" denotes that we remove the warping operation from our framework. "Ours w/o cross-frame attention" means that we replace cross-frame attention with self-attention modules in the networks. We make quantitative and qualitative comparison of the translated results. The quantitative comparison results are presented in the last three rows in Table 1. As shown, "Ours w/o latent alignment" and "Ours w/o cross-frame attention" both suffer from obvious performance drop in terms of "CLIP-Image" and "Warp Error". This indicates that it is effective for enhancing temporal coherency to constrain the *query*, *key*, and *value* tokens in the attention operation. We further ablate the visualization effect of latent alignment. The results is illustrated in Fig. 2 and 5. It could be obviously observed that with the help of latent alignment, the location of randomly generated shiny stars and city lights in the background is constrained throughout the sequence, effectively alleviating the temporal inconsistency issue.

## 6 Limitations and Discussions

**Limitations**. Although our method demonstrates impressive results on translating realistic videos, there still remains room for improvements. Firstly, the framework is designed for video translation tasks, where the structure and outline are not changed before and after translation. In contrast, video editing tasks which require shape deformation are beyond its capacity. Because optical flow maps extracted from the original video may not reflect the semantic correspondence in the generated video. Secondly, the performance of our proposed framework is subjected to the accuracy of optical flow estimation. If assisted with progressive optical flow estimation techniques, our method could achieve better results.

**Discussions**. Recently, several methods have been proposed to address the temporal coherence issue in diffusion based video translation tasks, including VideoControlNet Hu & Xu (2023), Rerender-A-Video Yang et al. (2023), and TokenFlow Geyer et al. (2023). VideoControlNet and Rerender-A-Video also leverage optical flow maps to transfer the motion information from the original video to the generated video. However, they focus on pixel-level temporal consistency by introducing pixel-aware cross frame latent fusion or performing inpainting on the occluded pixel regions. Therefore, a problem that these methods cannot avoid is the information loss introduced by the autoencoder. The encoding and decoding process would introduce color bias, which could be easily accumulated in the generated sequence. To tackle the problem, Yang et al. (2023) introduces a loss estimation method called fidelity-oriented image encoding, making the overall framework redundant. In our proposed framework, all the operations are conducted on the latent space, avoiding the information loss caused by recursive encoding and decoding process. TokenFlow also aims to enhance temporal consistency by ensuring the consistency in the diffusion feature space. In comparison, there are two main differences between TokenFlow and our approach. Firstly, TokenFlow captures inter-frame semantic correspondences by calculating cosine similarity between token feature. Instead, we take advantage of the off-the-shelf optical flow estimation methods to achieve the same goal. Secondly, TokenFlow propagates the token features, *i.e.* the attention output, to maintain temporal consistency. Differently, we apply warping operation in the latent space to constrain the *query* tokens.

## 7 Conclusion

In this paper, we propose a new zero-shot video-to-video translation framework, named *LatentWarp*. To enforce temporal consistency of *query* tokens, we incorporate a warping operation in the latent space. During the denoising process, we utilize optical flow maps to warp the latents of the last frame, aligning them with the current frame. This results in closely-related *query* tokens and attention outputs across corresponding regions of adjacent frames, improving latent-level consistency and enhancing the visual temporal coherence of generated videos. We validate the effectiveness of our framework through experiments on realistic videos.

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

# 8 APPENDIX

## 8.1 OVERALL ALGORITHM

---

**Algorithm 1:** *LatentWarp* zero-shot video-to-video translation

---

**Input:** Original video $\mathcal{I}$, a target prompt $\mathcal{P}$;
**Hyper-parameters:** Denoising steps $T$;
**Output:** Translated video $\hat{\mathcal{I}}$;
Perform opyical flow estimation and get optical flow maps $\mathcal{BM}$ and $\mathcal{FM}$;
# Translate the first frame :
Translate $I^1$ to $\hat{I}^1$ and save latents $\hat{\mathcal{Z}}^1$;
# Translate the other frames :
**for** $i = 2, 3, \ldots, n$ **do**
    **for** $t = T, T-1, \ldots, 1$ **do**
        # Warp the latent in denoising steps :
        **if** $t > T_0$ **then**
            According to Eq. 7, update $\hat{z}^i_{t-1}$;
        **end**
        # In the last few steps :
        **else**
            Following regular denoising process, update $\hat{z}^i_{t-1}$;
        **end**
    **end**
    Decode latent $\hat{z}^i$ to get the $i$-th translated frame $\hat{\mathcal{I}}^i$;
**end**

---

