# OpenReview forum: "LatentWarp: Consistent Diffusion Latents for Zero-Shot Video-to-Video Translation"
_ICLR.cc/2024/Conference — Submitted to ICLR 2024_

### Official Review · Reviewer_9TQZ · 2023-10-29

**Soundness:** 2 fair
**Presentation:** 2 fair
**Contribution:** 2 fair
**Rating:** 6
**Confidence:** 5

**Summary:**

This paper introduces LatentWarp, a framework for zero-shot video-to-video translation using image diffusion models. It addresses the challenge of maintaining temporal consistency in generated video frames. LatentWarp focuses on constraining query tokens for temporal consistency. It achieves this by warping the latent features from the previous frame to align with the current frame using optical flow information.  Extensive experiments confirm the superiority of LatentWarp in achieving high-quality video-to-video translation with temporal coherence.

**Strengths:**

The introduction of the LatentWarp framework offers a novel approach to zero-shot video-to-video translation. LatentWarp  emphasizes on preserving temporal consistency during video frame generation, achieved through optical flow and warping operations, significantly enhances temporal coherence,  which is a crucial aspect of video generation. The writing is good, and the structure of the paper is clear.

**Weaknesses:**

I find this method to be quite intuitive. My concerns mainly pertain to the experimental aspects:

1. Data-related issues: The authors do not compare their method to datasets used in their previous work like "tune-a-video." This omission may undermine the fairness of the experimental results.

2. Base model choice: The authors employ ControlNet as the base model instead of using LDM directly. ControlNet offers strong structural control, which might make the improvement from LatentWarp seem relatively small. It would be beneficial to provide experimental configurations with LatentWarp combined with LDM or Tune-a-Video to showcase this point.

3. Quantitative data details: The authors seem to have omitted reporting the sizes of the datasets used in their quantitative experiments.

4. Compared methods: It is advisable for the authors to compare their method with a broader range of existing approaches, such as Video-P2P and more recent methods.

5. Supplementary material: The authors have not provided corresponding video supplementary materials to visually assess temporal consistency.

6. User surveys: The authors did not provide user surveys as prior works have done. This is important to evaluate the visual effect.

7. Running costs: Editing time and GPU resource consumption, should be reported and compared to help readers understand the resource requirements and efficiency.

These concerns should be addressed to enhance the completeness and rigor of the experimental evaluation in the study.

**Questions:**

See weaknesses. I will rejudge the rating according to the rebuttal.

---

> ### Author Response · Authors · 2023-11-19
> **Authors' response (1/2)**
>
> Thank you for the supportive feedback.
>
> 1. **Data-related issues**
>
>    In Section 5.1 of our paper, we have claimed that the dataset used for our experiments is DAVIS, which is the same dataset used in "Tune-A-Video." We acknowledge that in some cases, such as in Figure 1, we choose to include partial images which is not from DAVIS. Our intention is to demonstrate the performance of our approach under challenging conditions and highlight its effectiveness on handling complex objects and  backgrounds.
>
> 2. **Base model choice**
>
>    Firstly, it is important to note that many of the comparative methods we evaluated in our study also utilized ControlNet, such as Rerender a video. This choice was made to ensure a fair and consistent baseline comparison across different techniques. However, we understand the concern that ControlNet's strong structural control may potentially diminish the observed improvement from LatentWarp. To address this concern, we have conducted experiments comparing the performance of LatentWarp with and without controlnet in the ablation experimentaion in the [supplementary material](https://diffusion-latentwarp.github.io/sm/supp.html). Such experiments demonstrate the distinct contributions and improvements offered by LatentWarp using LDM directly.
>
> 3. **Quantitative data details**
>
>    Our evaluation dataset comprises of 24 text-video pairs from DAVIS, i.e., twelve videos and each video corresponds to two prompts. We extract the first 24 frames of each video and calculate the metrics.
>
>    The twelve videos are bear, blackswan, boat, bus, camel, car-turn, dog-agility, elephant, flamingo, gold-fish, hike, and train.
>
>    We will supplement these details in our revised paper.
>
>
>
> 4. **Compared methods**
>
>    We conducted extensive experiments to demonstrate the effectiveness and superiority of our proposed method in comparison to more recent approaches,  including Rerender a video, Tokenflow, Video-P2P, and others. The detailed results of these video-based comparisons can be found in in the [supplementary material](https://diffusion-latentwarp.github.io/sm/supp.html).
>
> 5. **Supplementary material**
>
>    Please see the corresponding video [supplementary materials](https://diffusion-latentwarp.github.io/sm/supp.html)

---

> ### Author Response · Authors · 2023-11-19
> **Authors' response (2/2)**
>
> 6. **User surveys:**
>
>    Good suggestion. We refer to previous works[1,2,3] and adopt a Two-alternative Forced Choice (2AFC) protocol, where participants are shown the original video, the translation result of a compared baseline method and the result of our method. The participants are asked to choose which result they prefer.
>
>    We select 15 video-prompt pairs for evaluation and we collect 420 votes from 28 users. The results are provided in the table below, which also verifies the superiority of our method considering realistic user evaluations. We will include this result in our revised paper.
>
>    | Methods                       | Tune-A-Video | Text2Video-Zero | TokenFlow |
>    | ----------------------------- | ------------ | --------------- | --------- |
>    | User preference of our method | 88.6%        | 80.0%           | 72.9%     |
>
> 7. **Running costs**
>
>    We have measured the inference time and GPU resource consumption of different models for reference. Because the number of generated frames  are limited for some methods, for fair comparison,  the video length is set to be 8 and the image size is set as 512 x 512 for all the methods. 50-step DDIM sampling strategy is adopted. All the experiments are conducted on one NVIDIA A100 GPU. From the table, it can be seen that our method is efficient concerning both the inference time and GPU memory consumption.
>
>    This could be attributed to reasons of two aspects.  The main advantage of our method is that our method skips the DDIM inversion process, which is typically adopted in video editing tasks to preserve the temporal prior while it consumes lots of time. However, the objective could be achieved through latent alignment operation alternatively, which is obviously more efficient. Moreover, since the warping operation is performed on the latent space, a batch of frames could be denoised on parallel. In contrast, Rerender-A-Video perform the warping operation on the pixel level, which makes the frames need to be generated one by one.
>
> **Inference time.**
>
> | methods          | flow extraction | tuning | inversion | sampling | total |
> | ---------------- | --------------- | ------ | --------- | ------- | ----- |
> | video p2p        | -               | 1200s  | 720s      | 55s     | 1975s |
> | tune a video     | -               | 1200s  | 8s        | 13s     | 1221s |
> | tokenflow        | -               | -      | 55s       | 16s     | 71s   |
> | text2video-zero  | -               | -      | -         | 18s     | 18s   |
> | rerender a video | 20s             | -      | -         | 182s    | 202s  |
> | Ours             | 18s             | -      | -         | 23s     | 41s   |
>
> **GPU memory consumption.**
>
> | methods          | flow extraction | tuning | inversion | sampling | max  |
> | ---------------- | --------------- | ------ | --------- | ------- | ---- |
> | video p2p        | -               | 10G    | 29G       | 29G     | 29G  |
> | tune a video     | -               | 10G    | 29G       | 29G     | 29G  |
> | text2video-zero  | -               | -      | -         | 31G     | 31G  |
> | tokenflow        | -               | -      | 14G       | 14G     | 14G  |
> | rerender a video | 3G              | -      | -         | 13G     | 13G  |
> | Ours             | 2G              | -      | -         | 16G     | 16G  |
>
> [1] Geyer, Michal, et al. "Tokenflow: Consistent diffusion features for consistent video editing." *arXiv preprint arXiv:2307.10373* (2023).
>
> [2] Kolkin, Nicholas, Jason Salavon, and Gregory Shakhnarovich. "Style transfer by relaxed optimal transport and self-similarity." *Proceedings of the IEEE/CVF Conference on Computer Vision and Pattern Recognition*. 2019.
>
> [3] Park, Taesung, et al. "Swapping autoencoder for deep image manipulation." *Advances in Neural Information Processing Systems* 33 (2020): 7198-7211.

---

### Official Review · Reviewer_cpKU · 2023-10-31

**Soundness:** 3 good
**Presentation:** 3 good
**Contribution:** 3 good
**Rating:** 6
**Confidence:** 4

**Summary:**

In this paper, the author studies the task of zero-shot video editing and addresses the problem of temporal consistency for the edited videos. The author points out that existing methods only consider the K/V tokens in the cross-attention mechanism and ignore the Q token which plays a more important role in preserving temporal consistency. Specifically, the author proposes to adopt optical-flow to warp the latent feature from the previous frame. The overall idea is interesting and the experimental results look good.

**Strengths:**

1. The idea of considering the consistency of query tokens in cross-attention to generate consistent videos is interesting.
2. The writing is clear and easy to follow and the experimental results are promising.

**Weaknesses:**

1. In section 5.1, the overall denoising step number is set to 20 and the proposed method is only applied to the first 16 steps. It would be good if there could be an ablation study about the two stages of the denoising steps.
2. Is the proposed method sensitive to the hyper-parameters \alpha and threshold as well as the optical-flow methods? I would like to see some ablation studies on that.
3. Is it possible to edit/add some specific object to the video? Like adding a hat on a running dog? It seems most of the cases shown in the paper are about style changes. I would like to see some complex cases.

**Questions:**

Please refer to the weakness part.

---

> ### Author Response · Authors · 2023-11-19
>
> Thank you for the supportive feedback.
>
> 1. **Ablation studies on performing latent alignment on the last few denoising steps.**
>
> ​    We conduct the experiment of applying latent alignment operation on the first 16 steps and on the last 4 steps in the denoising process, and present the video results in the [supplementary material](https://diffusion-latentwarp.github.io/sm/supp.html#Ablations).  Although the visual consistency can also be maintained through the last few steps, we observe that errors in the warping operation and optical flow estimation would be introduced in the denoising process especially in the last few steps and reflected on the generation result, hampering the generation quality.
>
> 2. **Ablation studies on hyper-parameter tuning and optical flow network choice.**
>
>    Please refer to the global response for abltion on hyper-parameter. We also provide the video results with the optical flow extracted by GMFlow[1] and there are no obvious differences.
>
> 3. **Editing with structure deviations.**
>       1. In this paper, we deal with a specific video translation task, preserving temporal consistency of stylized videos with original videos in terms of both object appearance and motion is a primary objective of our method. Thus, our method may handle some specific structure deviations which to an extent can be viewed as a by-product of stylization.
>       2. More general video editing, like adding a hat on a running dog, involves significant object structure deviations, and usually requires a motion prior to ensure a consistent editing across frames.  The challenging task exceeds the focus of this paper, and we remain it as a direction to extend our work in the future.
>
> [1] Xu H, Zhang J, Cai J, et al. Gmflow: Learning optical flow via global matching[C]//Proceedings of the IEEE/CVF conference on computer vision and pattern recognition. 2022: 8121-8130.

---

### Official Review · Reviewer_WQHp · 2023-10-31

**Soundness:** 2 fair
**Presentation:** 2 fair
**Contribution:** 2 fair
**Rating:** 5
**Confidence:** 5

**Summary:**

This paper introduces optical flow for video-to-video translation by warping the latent codes in diffusion’s sampling process and achieves the SOTA performance on V2V.

**Strengths:**

1. The motivation is well presented,  the analysis of constrained query tokens would lead to consistent output, making the choice of warping the latent code convinced.

2. The method is straightforward, and the results seem good.

**Weaknesses:**

1. Introducing the optical flow into diffusion-based video processing has been studied by Rerender-A-Video, though it is not applied to the latent.
2. While this is video processing work, there are no video results, which makes it hard to distinguish the visual quality.
3. Authors introduce the occlusion mask for indicating the warped region and unwrapped region, but how to guarantee the consistency on the unwrapped region?
4. Some related works are not compared, such as Edit a Video, Rerender-A-Video, and etal.

**Questions:**

please present the video comparisons.

---

> ### Author Response · Authors · 2023-11-19
>
> Thank you for the supportive feedback.
>
> 1. **Comparison with Rerender-A-Video.**
>
>    Although introducing optical flow into diffusion-based video processing has been explored in prior work, such as Rerender-A-Video, there are several key differences in our approach:
>
>    1. Firstly, we extract both forward optical flow maps and backward optical flow maps between adjacent frames and apply a  warping operation on the latent of the last frame to align with the current frame using these flow maps.
>    2. The main way we calculate the binary mask for preserving or replacing parts of the warped latent differs from previous optical-flow based diffusion methods, such as Rerender-A-Video. Our methods utilizes a binary mask to determine which regions of the warped latent should be preserved and which should be replaced with latent of the current frame. The details is illustrated in method 4.3.
>    3. There are also differences in the fusion stage between our method and Rerender-A-Video. This stage involves combining the warped latents with the generated latents to produce the final output.
>
> 2. **Presentation of video results.**
>
> ​     Please see our global response.
>
> 3. **How to keep the consistency of the unwarped regions.**
>
> ​      We preserve the consistency of the unwarped regions through cross-frame attention mechanism, i.e., sharing the *key* and *value* tokens across attentions of different frames, to encourage the temporal consistency. As illustrated in the [ablation study, ](https://diffusion-latentwarp.github.io/sm/supp.html#Ablations) without cross-frame attention, the left part of the scene exposed to us with the camera turning left would appears significantly inconsistent with the other regions, hampering the video quality, while our method effectively avoids this issue.
>
> 4. **Comparison with Edit-A-Video and Rerender-A-Video.**
>
> ​      Since Edit-A-Video has not released the code so far, we provide the comparison with Rerender-A-Video and other methods in the [supplementary material](https://diffusion-latentwarp.github.io/sm/supp.html). We will cite and discuss these methods in our revised paper.

---

### Official Review · Reviewer_Yr5G · 2023-10-31

**Soundness:** 3 good
**Presentation:** 3 good
**Contribution:** 2 fair
**Rating:** 5
**Confidence:** 4

**Summary:**

The paper employs a straightforward strategy to maintain temporal consistency in query tokens by introducing a warping operation in the latent space. This operation aligns the generated latent features of the previous frame with the current one during the denoising process, utilizing optical flow from the original video. As a consequence, adjacent frames share closely-related query tokens and attention outputs, fostering latent-level consistency and thereby enhancing the visual temporal coherence of the generated videos. Extensive experimental results underscore the effectiveness of LatentWarp in accomplishing video-to-video translation while preserving temporal coherence.

**Strengths:**

1. The propose method is well-motivated.
2. The paper is well-structured, capable of clearly elucidating its core ideas.
3. The conducted experiments adequately showcase the efficacy of the method being proposed.

**Weaknesses:**

1. It is hard for me to see the improvement of temporal consistence from the images. Therefore, it is strongly advised to incorporate the video in the supplementary materials.

2. Section 4.2.  r^{i}|wrap(I^{i-1}, m^{i->i-1})-I^{i}| should be r^{i}|wrap(I^{i-1}, m^{i-1>i})-I^{i}| ?

3. I understand  that warped query can enhance the temporal consistence but why does it improve visual details(Figure 5) as well?

4. Since there are lots of open-sourced video diffusion models, which can naturally ensure the temporal consistence, what's the benefit of using Image diffusion model for video editing?   Longer video?

5. For different video, the hyperparameters of Eq.(6) should be selected differently?

**Questions:**

Please see the weakness.

---

> ### Author Response · Authors · 2023-11-19
>
> Thank you for the supportive feedback.
>
> 1. **Presentation of video results**
>
>    Please see our global response.
>
> 2. **Explanation of the equation in section4.2.**
>
>    It is not a mistake.  $warp(I^{i-1}, m^{i->i-1})$, which is used in our paper, refers to a backward warping operation on the (i-1)-th frame $I^{i-1}$ with the backward optical flow .  Differently, $warp(I^{i-1}, m^{i-1->i})$ denotes a forward warping operation with forward optical flow. Both of them can warp the last frame to the current one. However, backward warping has some advantages over forward warping. One of the main advantages is that it can avoid the problem of holes, which is common in forward warping. Holes refer to situations where some pixel points do not have corresponding original pixel points after mapping, resulting in empty spaces or non-existent pixels, which is illustrated in [this link](https://www.researchgate.net/figure/Forward-and-backward-image-warping-In-the-case-of-foward-warping-A-holes-can-occur_fig2_267946997). Therefore, we incorporate backward warping in our method rather than forward warping.
>
> 3. **Clarification of Fig. 5**
>
>    Sorry for this misunderstanding. In Figure 5, we aim not to show the improvement of our method, but to demonstrate that latent alignment effectively preserves the fine-grained visual details. We will clarify this in our revised paper.
>
> 4. **Advantages of zero-shot video translation techniques compared with video diffusion model.**
>
>    Compared with video diffusion models, our zero-shot video translation technique has the following advantages:
>
>    1) One significant advantage is the ability to handle longer videos. Image diffusion models typically operate on individual frames, allowing them to handle videos of arbitrary length without the computational and memory limitations that may arise with video diffusion models.
>
>    2) Furthermore, image diffusion models provide flexibility in the editing process. They allow for more fine-grained control over individual frames, enabling targeted modifications and adjustments to specific parts of the video without affecting the entire sequence. This can be particularly useful in scenarios where precise video editing is required, such as in film production or content creation.
>
> 5. **Hyper-parameter selection.**
>
>    Please see our global response.

---

### Author Response · Authors · 2023-11-19
**Global Response**

We sincerely thank the reviewers for their kind feedbacks.

We would like to address several common points raised by all reviewers in this response.

**The presentations of video results.**

Thank you for your suggestions. We have provided a supplementary material in this [link](https://diffusion-latentwarp.github.io/sm/supp.html), containing the presentation of our method's results and video comparisons with recent methods. It could be observed that our video results are impressive and competitive. We further complement the ablation results in the supplementary material, which demonstrate the effectiveness and generality of our method.

**Ablations on hyper-parameters**

All the videos presented in our [supplementary material](https://diffusion-latentwarp.github.io/sm/supp.html#Ablations) are generated with the same hyper-parameters. As indicated in our paper, $\alpha$ is set as 5 and threshold is set as 0.6. Moreover, we provide an ablation study on the choice of hyper-parameter in the supplementary material. With $\alpha$ ranging from 4 to 6 and threshold ranging from 0.5 to 0.7, the video results show similar outcomes. This demonstrates that our method is robust to hyper-parameter choice.

---

### Author Response · Authors · 2023-11-22
**Sincerely Looking Forward to Your Feedback**

**Dear Reviewers,**

We sincerely appreciate your great efforts in reviewing this paper. Your constructive advice and valuable comments have significantly improved our work. Given the approaching deadline, we would be grateful if you could let us know if you have any further concerns. We hope you will take our responses into consideration during your assessment, and we are always available to further address any remaining concerns or unclear explanations.

Once again, we deeply appreciate the time and effort you have dedicated to our paper.

Best regards,

Paper 936 Authors

---

### Meta-Review · Area_Chair_sw1V · 2023-12-11

**Metareview:**

This paper addressed zero-shot video editing by warping latent maps to enforce temporal consistency.

All the reviewers are within the borderlines with two rating the paper marginally below and two above the acceptance threshold.

The main concerns were:
- missing video results
- unclear advantages over video diffusion models.
- lack of user study.

The initial paper did not provide video results in the supplementary material. During the rebuttal, it has been provided via an external website.

The video results comparison with others are mixed.
- Overall, the successful examples shown on the web are global style transfer.
- The advantages over TokenFlow and Re-render a video, and Gen 1 are not clear.
- Why is TokenFlow's results have a weird unedited frame? It's inconsistent with the results reported in their website.
- In "a fluffy wolf doll", the editing of proposed method did not respect the instruction.

Unfortunately, all the reviewers did not engage in the discussions. The AE believes evaluating these additional results (which the authors omitted during the submission) will likely take another round of reviews to fully evaluate the proposed method. The AE thus recommends to reject.

**Justification For Why Not Higher Score:**

The authors did not provide visual evaluation in the supplementary material. They provided it via an external website during the rebuttal. However, the AE finds the results not sufficiently convincing and the results may need to go through another round of reviews by the reviewers.

**Justification For Why Not Lower Score:**

N/A

---

### Decision · Program_Chairs · 2024-01-16

Reject